# Hsa_circ_0056686, derived from cancer-associated fibroblasts, promotes cell proliferation and suppresses apoptosis in uterine leiomyoma through inhibiting endoplasmic reticulum stress

**Meifang Suo[1], Zhichen Lin[2], Dongfang Guo[1], Airong Zhang[1] ***

**1** Department of Medical Laboratory, Zhumadian Central Hospital, Zhumadian, China, **2** Department of Burns, the 990th Hospital of the Joint Staff of the People's Liberation Army, Zhumadian, China

* airongz@21cn.com

**Data Availability Statement:** All relevant data are within the paper and its Supporting Information files.

## Abstract

Abnormal expression of circular RNAs (circRNAs) in cancer-associated fibroblasts (CAFs) is involved in the tumor-promoting ability of CAFs. Hsa_ circ_ 0056686 has been reported to affect leiomyoma size. The purpose of this study is to investigate the regulatory role of hsa_circ_0056686 in CAFs on uterine leiomyoma (ULM). The primary CAFs and corresponding normal fibroblasts (NFs) were isolated from the tumor zones of ULM tissues and adjacent, respectively. Hsa_circ_0056686 level was higher in CAFs than NFs, and also higher in ULM tissues than in adjacent tissues. CAFs-CM significantly increased the proliferation and migration and inhibited apoptosis of ULM cells, as confirmed by CCK-8, transwell, and flow cytometry assays. Moreover, conditioned medium (CM) from CAFs transfected with hsa_circ_0056686 shRNA (CAFs^sh-circ_0056686-CM) abolished CAFs-mediated proliferation, migration and apoptosis of ULM cells. CAFs-CM suppressed the expression of endoplasmic reticulum stress (ERS) marker proteins and induced the expression of extracellular matrix (ECM) marker proteins, thus suppressing ERS and increasing ECM accumulation, which could be declined by CAFs^sh-circ_0056686-CM. Meanwhile, knockdown of hsa_circ_0056686 reversed the inhibitory effects of CAFs-CM on brefeldin A-induced cell apoptosis. Luciferase gene reporter and RNA pull-down assays indicated that miR-515-5p directly bound with hsa_circ_0056686. MiR-515-5p overexpression restored the hsa_circ_0056686-shRNA-mediated malignant biological behaviors of ULM cells. Hsa_circ_0056686 contributed to tumor-promoting effects of CAFs in ULM, manifested by promoting ULM cell proliferation and migration and reducing ERS-induced apoptosis through sponging miR-515-5p.

**Funding:** The author(s) received no specific funding for this work.

**Competing interests:** The authors have declared that no competing interests exist.

## Introduction

As the most common benign tumor in reproductive women, uterine leiomyoma (ULM) has become an important public health issue. It is reported that ULM is the indication for all hysterectomies performed. This common ULM, due to its invasion and metastasis different from other malignant tumors, brings a variety of harm to women, such as menstrual abnormalities, recurrent pregnancy loss, gynecological diseases, pelvic pressure and so on [1]. Although the occurrence and development of leiomyoma have been deeply studied, the etiology and pathogenesis of leiomyoma remain unclear. Therefore, the molecular mechanisms and new treatments of ULM have become research hotspots.

Current study has reported that solid tumors are truly heterogeneous tissues, and tumor progression and metastasis are instigated by the bidirectional communication between tumor cells and tumor micro-environment (TME), rather than by cancer cells alone. Cancer-associated fibroblasts (CAFs) are the most important stromal cell types in TME. They are considered to be the accomplices of tumor progression and play an important role in tumor growth, progression, metastasis, angiogenesis and immune responses [2,3]. CAFs could secrete a variety of factors including chemokines, cytokines, exosomes and growth factors to affect cancer cell fate [4,5]. For instance, CAFs promte tumorigenesis of colorectal cancer by the secretion of exosomal circSLC7A6 [6]. However, it is still largely unclear about the exact function of circRNAs in CAFs of ULM.

Circular RNAs (circRNAs) are a new type of endogenous non-coding RNA, which are RNA molecules with covalently joined 3'- and 5'- ends formed by back-splice events, thus presenting as closed continuous loops, which makes them highly stable [7]. Accumulated evidence showed that circRNAs are abnormally expressed in many cancers including ULM, which are key players in cancer initiation, development and progression. Specially, in the pathological processes of tumor cells, circRNAs participate in cell proliferation, apoptosis, invasion and migration [8,9]. For example, hsa_circ_0043265 level is significantly upregulated in non-small lung cancer cells, and can act as an oncogene to restrain proliferation, migration, invasion and EMT process in non-small cell lung cancer [10]. Hsa_circRNA_101036 is downregulated in oral squamous cell carcinoma cells, and functions as a tumor suppressor via inducing endoplasmic reticulum stress (ERS) in oral squamous cell carcinoma cells [11]. Wang et al. found that hsa_circ_0056686 is highly expressed in ULM [12], but its role and molecular mechanism need to be further explored.

In the present study, we aim to explore the role of hsa_circ_0056686 derived from CAFs in ULM. The results of this study revealed that the expression of hsa_circ_0056686 was upregulated in CAFs and ULM tissues, and could regulate proliferation, migration, ECM accumulation, ERS induced-apoptosis by sponging miR-515-5p.

## Material and methods

### Sample collection

A total of 30 cases of ULM tissues and adjacent myometrial tissues were collected from patients undergoing myomectomy or total hysterectomy at Zhumadian Central hospital from May 2016 to June 2018. The mean age of patients was $42 \pm 0.6$ years with a range of 32 to 54 years, and they did not take hormonal treatments for at least 3 months before hysterectomy. The menstrual cycle among patients were identified in the follicular phase or in the luteal phase and diagnoses of leiomyoma were clear. The surgery was performed 3–10 days after cessation of menstruation. Written informed consent was obtained from all patients before operation. Tissues were stored in liquid nitrogen, and the histopathologic diagnoses were all confirmed as

ULM and without degeneration. The protocol of this study was approved by the ethics committee of Zhumadian Central hospital, and written informed consent was obtained from each participant (ZMDCH-2016-021).

## Cell culture

The leiomyoma smooth muscle cells (LSMCs) were isolated as previously described [13]. Tissue samples were washed with phosphate-buffered saline (PBS), cut into small pieces and centrifuged at l000 rpm for 5 min. The supernatant was discarded. After that, Dulbecco's modified Eagle medium (DMEM) (Gibco, Rockville, MD, USA) containing type I collagenase was added for digestion via thermostatic water bath at 37˚C for 4–6 h, and the mixture was filtered by using a 300-mesh (with a mesh diameter of 38 μm) stainless steel cell filter screen. The filtrate was centrifuged at l000 rpm for 5 min, and the supernatant was removed. The sediment was suspended by using an appropriate amount of DMEM culture solution containing 10% fetal bovine serum (FBS) and incubated in a humidified atmosphere containing 5% $CO_2$ at 37˚C. After adhering to the wall overnight, cells were observed microscopically. The culture flask was shaken slightly, and the culture solution was fully replaced to remove other extraneous cells that did not adhere to the wall. After that, the culture solution was replaced every 3 days.

The NF and CAF populations were successfully isolated from the normal and tumor zones of uterine leiomyoma tissues of the same patient as previously described [14]. The primary fibroblasts were cultured in DMEM-high glucose with 10% FBS in a humidified incubator containing 5% $CO_2$ at 37˚C. The cells were incubated until they were 70% confluent, washed with phosphate-buffered saline (PBS) three times, and finally cultured in serum-free medium for another 48 h to prepare conditioned medium (CM). The CM was collected and centrifuged for 10 min at 3000 rpm to remove cell debris. CM was cryopreserved at -80˚C. All fibroblasts used for in vitro study were 3–10 passages.

## Cell transfection

Hsa_circ_0056686 short hairpin RNA (shRNA) (sh-hsa_circ_0056686) or its negative control (sh-NC), miR-515-5p mimic and miR-515-5p inhibitor and their negative controls (NC mimic and NC inhibitor) were synthesized by Ribobio (Guangzhou, China). Cells were transfected with oligonucleotides or their control by using lipofectamine 2000 (Invitrogen, CA, USA) at a final concentration of 50 nM following the manufacturer's instructions and then incubated for 48 h.

## Quantitative real-time PCR (RT-qPCR) analysis

Total RNA was extracted by using Trizol (Invirtogen, Carlsbad, CA, USA). The cDNA was synthesized from total RNA by using Prime Script RT reagent (TaKaRa, Tokyo, Japan). RT-qPCR was performed with SYBR Green PCR Master Mix (Invitrogen, USA) and an Agilent Bioanalyzer 2100 (Agilent Technologies, Santa Clara, CA, US). GAPDH was used as endogenous controls. Briefly, 2 μL of cDNA was added to 10 μL of the 2× SYBR green PCR master mix with 0.4 μL of Taq polymerase enzyme (RiboBio, China), 0.8 μL of each primer and 6 μL ddH2O to a final volume of 20 μL. The qPCR cycling conditions consisted of: 95˚C for 2 min; then 35 cycle amplification for 20 s at 95˚C, 30 s at 55˚C, 15 s at 72˚C; followed by 1 min at 72˚C. The primers used in this study were synthesized by Sangon Biotech (Shanghai, China). All reactions were run in triplicate, and the relative gene expression was calculated with the $2^{-\Delta\Delta CT}$ method.

## Cell proliferation

Cell growth was assessed by Cell Counting Kit-8 (CCK-8) assay. LSMCs were plated at $1 \times 10^3$ cells per well into 96-well plates with six replicate wells at the indicated concentrations in a humidified atmosphere containing 5% $CO_2$ at 37˚C. After 24, 48, 72 and 96 h incubation, CCK-8 solution (10 μL) was added to each well and further incubated for 4 h. The absorbance was measured at 450 nm by using microplate reader (Molecular devices, Shanghai, China).

## Flow cytometry analysis

The LSMCs were cultured for 48 h, and then digested with trypsin. After centrifugation, cells were harvested and re-suspended with binding buffer. Then, Annexin V-FITC/PI apoptosis detection kit (BD Bioscience, San Jose, CA, USA) was used to detect the cell apoptosis. Briefly, cells were stained with 5 μL of Annexin V-fluorescein isothiocyanate (V-FITC) and 10 μL of Propidium iodide (PI) for 15 min in dark. Fluorescence signals were analyzed directly by flow cytometry (BD Bioscience, San Jose, CA, USA) by using the Cell Quest program (Becton Dickinson, Franklin, NJ).

## Transwell migrated assay

Cell migration assay was performed with Matrigel-coated transwell chamber (Corning Incorporated, Corning, NY, USA). Briefly, the lower compartments were filled with DEME, and then pre-coated with 10 mg/mL collagen for 1 h at 37˚C. In each group, $5 \times 10^4$ cells were seeded into upper chamber with serum-free DEME. The migrating cells to the lower layer were fixed with 4% paraformaldehyde and stained with 0.1% crystal violet. Six fields per well of view were randomly selected to calculate the number of migration cells under microscope (Olympus, Tokyo, Japan).

## Western blot analysis

Protein homogenates from osteoarthritic chondrocytes were extracted as previously described. Briefly, the cells were lysed for 20 min on ice in ice-cold lysis buffer (Roche). The lysates were centrifuged at 12,000× g for 20 min at 4˚C to obtain a clear lysate. The protein content of each sample was determined by using the BCA Protein Assay Kit (Thermo Scientific). Then, equal amounts of proteins (15 μg/lane) were separated on a 12% sodium dodecyl sulfate polyacrylamide gel electrophoresis (SDS-PAGE) and transferred to polyvinylidenedifluoride (PVDF) membranes (Bio-Rad, Hercules, CA, USA). After blocking with 5% non-fat milk, the membranes were incubated with primary antibodies: GAPDH (1:2000, ab8245, Abcam), GRP78 (1:1000, ab21685, Abcam), CHOP (1:1000, ab11419, Abcam), ATF6 (1:1000, ab62576, Abcam), COL1A1 (1:1000, ab138492, Abcam), COL1A2 (1:1000, ab96723, Abcam), COL3A1 (1:1000, ab7778, Abcam), Bcl-2 (1:1000, ab32124, Abcam) and Cleaved-caspase3 (1:1000, ab2302, Abcam) overnight at room temperature, and then incubated with horseradish peroxidase (HRP)-labeled goat anti-rabbit lgG (1:1000, ab6721, Abcam) at room temperature for 1 h. The protein bands were visualized by using the Enhanced chemiluminescence reagents (Millipore, MA, USA). The expression of relative protein was obtained by the gray value ratio of the target protein to the internal reference GAPDH and analyzed with ImageJ software (National Institutes of Health, Bethesda, MA, USA).

## Luciferase gene reporter assay

The wild-type and mutant-type hsa_circ_0043265 (hsa_circ_0043265-wt/mut) reporter vectors containing the miR-515-5p target binding sites were constructed by GeneChem Company

(Shanghai). The HEK293T cells were co-transfected with hsa_circ_0043265-wt/mut and miR-515-5p mimic in a 24-well plate. After 48 h of transfection, the luciferase activities were measured by using Dual-Luciferase Reporter Assay System according to manufacturer's protocol (Promega).

## RNA pull-down

The bio-probe-NC, bio-miR-515-5p-wt and bio-miR-515-5p-mut were synthesized (Ribobio) and labeled by using Biotin RNA Labeling Mix (Roche) and Sp6 RNA polymerase (Roche). The magnetic beads coated with streptavidin (Invitrogen) were incubated with biotin-labelled RNA at room temperature to prepare magnetic beads coated with probes. Then, LSMCs were lysed by cell lysis buffer (Sigma) and cell lysate were collected and incubated with probe coated beads at 4˚C for 1 h. The bound precipitates were eluted, and analyzed by RT-qPCR to verify the relationship between hsa_circ_0056686 and miR-515-5p.

## Statistical analysis

All statistical analyses were performed by using the SPSS software (ver. 23.0; SPSS, Chicago, IL). The quantitative data derived from three independent experiments were expressed as mean ± SD. Comparisons between two groups were made by the Student's $t$-test. Data between multiple groups were performed with one-way analysis of variance (ANOVA) followed by post hoc analysis with LSD test. $P < 0.05$ was considered statistically significant.

## Results

### CAFs promoted proliferation and migration and inhibited apoptosis in LSMCs

To identify the NFs and CAFs, the expression of CAF biomarkers α-SMA and FAP in primary cultured CAF cells was examined. The results confirmed that the expression of α-SMA and FAP was remarkably increased in CAFs compared with NFs (Fig 1A). To determine the role of CAFs in ULM cell functions, LSMCs were cultured with CAFs-CM and NFs-CM. The results of CCK-8 assay showed that CAFs significantly promoted proliferation of LSMCs. Whereas treatment of LSMCs with NFs-CM had a moderately promoting effect on cell proliferation (Fig 1B). Through Transwell assay, we investigated the effect of CM form CAFs on the migration of LSMCs. The results revealed that CAFs-CM caused a marked increase in the number of migrated cells compared with NFs-CM (Fig 1C). Flow cytometry assay indicated that CAFs strikingly suppressed apoptosis of LSMCs. Taken together, the above results suggested that CAFs could facilitate proliferation and migration and suppress apoptosis of LSMCs.

### Knockdown of hsa_circ_0056686 in CAFs inhibited proliferation, migration and induced apoptosis in LSMCs

We collected 30 pairs of ULM tissues and adjacent myometrial tissues to evaluate the expression levels of hsa_circ_0056686 by qRT-PCR. The results showed that hsa_circ_0056686 was upregulated in ULM tissues compared with normal myometrial tissues (Fig 2A). Besides, to investigate whether CAFs affect ULM progression through upregulation of hsa_circ_0056686, we detected the expression of hsa_circ_0056686 in LSMCs treated with CAFs-CM or NFs-CM by RT-qPCR. The results revealed that hsa_circ_0056686 was significantly upregulated in CAFs-CM group than in paired NFs-CM (Fig 2B). We next transfected sh-hsa_circ_0056686 (CAFs[sh-circ_0056686]) or negative control (CAFs[sh-NC]) into CAFs to explore whether hsa_-circ_0056686 is involved in the tumor-promoting ability of CAFs. The results of RT-qPCR

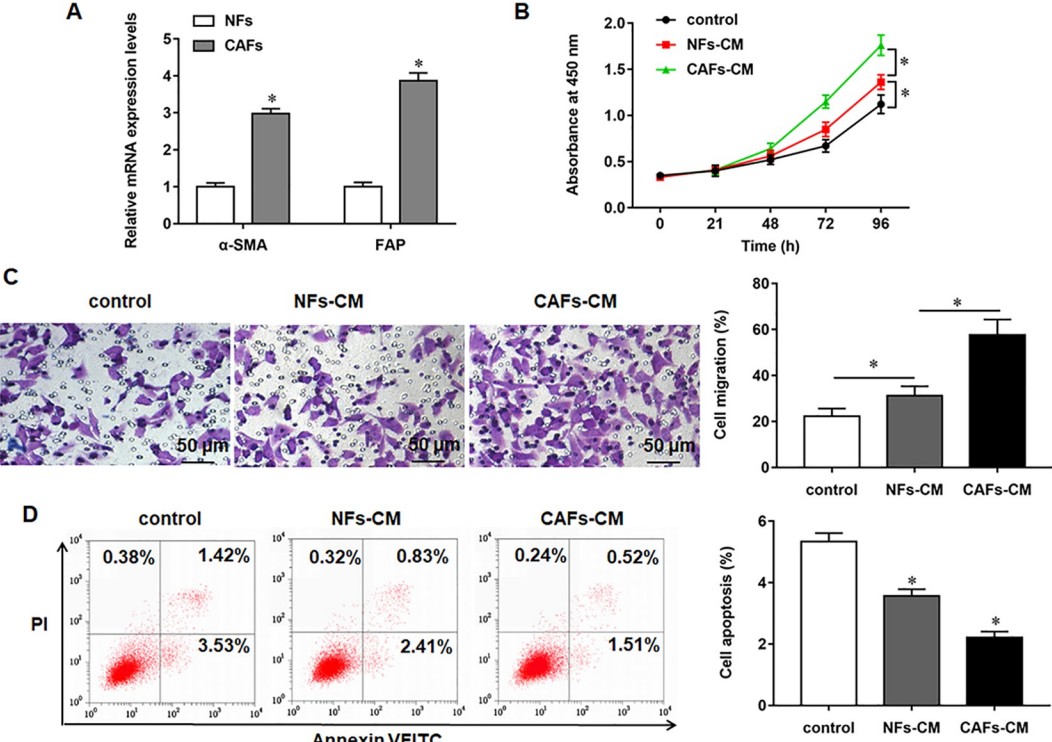

**Fig 1. Effect of CAFs on the proliferation, migration and apoptosis of LSMCs.** LSMCs were treated with CAFs-CM or NFs-CM. (A) Western blotting analysis of α-SMA and FAP expression in paired NFs and CAFs. (B) The proliferation ability of LSMCs was detected by CCK-8 assay. (C) The migration ability of LSMCs was detected by Transwell assay. (D) The apoptosis ratio of LSMCs was detected by flow cytometry assay. $^*P < 0.05$, $vs$ NFs (A), control (B-D).

suggested that the transfection of sh-hsa_circ_0056686 was efficient to downregulate hsa_-circ_0056686 (Fig 3A). CCK-8 and Transwell assays revealed that the promotion effects of CAFs-CM on the proliferation and migration of LSMCs were significantly reversed by CAFs$^{sh\text{-}circ\_0056686}$-CM (Fig 3B and 3C). Furthermore, it was verified by flow cytometry analysis that absence of hsa_circ_0056686 could countervail the inhibitory effect of CAFs on cell apoptosis (Fig 3D).

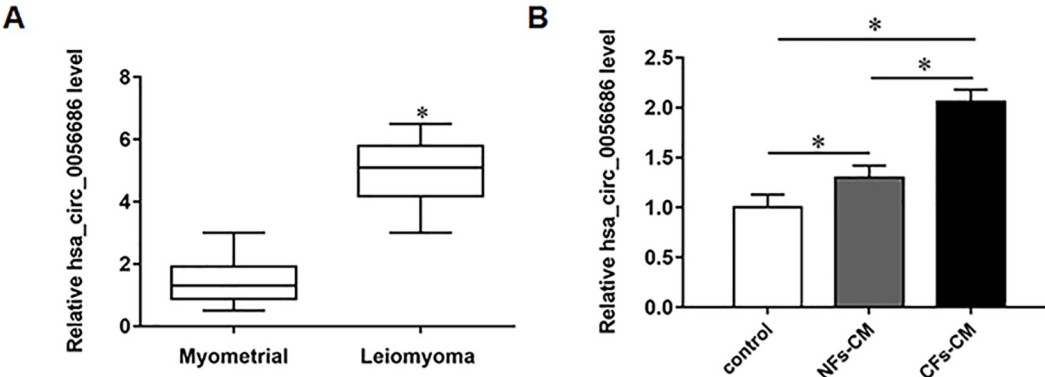

**Fig 2. The expression of hsa_circ_0056686 was upregulated in ULM tissues.** (A) (n = 30) Relative hsa_circ_0056686 expression level in myometrial tissues and leiomyoma tissues. (B) CAFs induced significant upregulation of hsa_circ_0056686 in CAFs-CM-treated LSMCs compared with control and NFs-CM groups. $^*P < 0.05$, $vs$ normal myometrial tissues (A), control and NFs-CM groups (B).

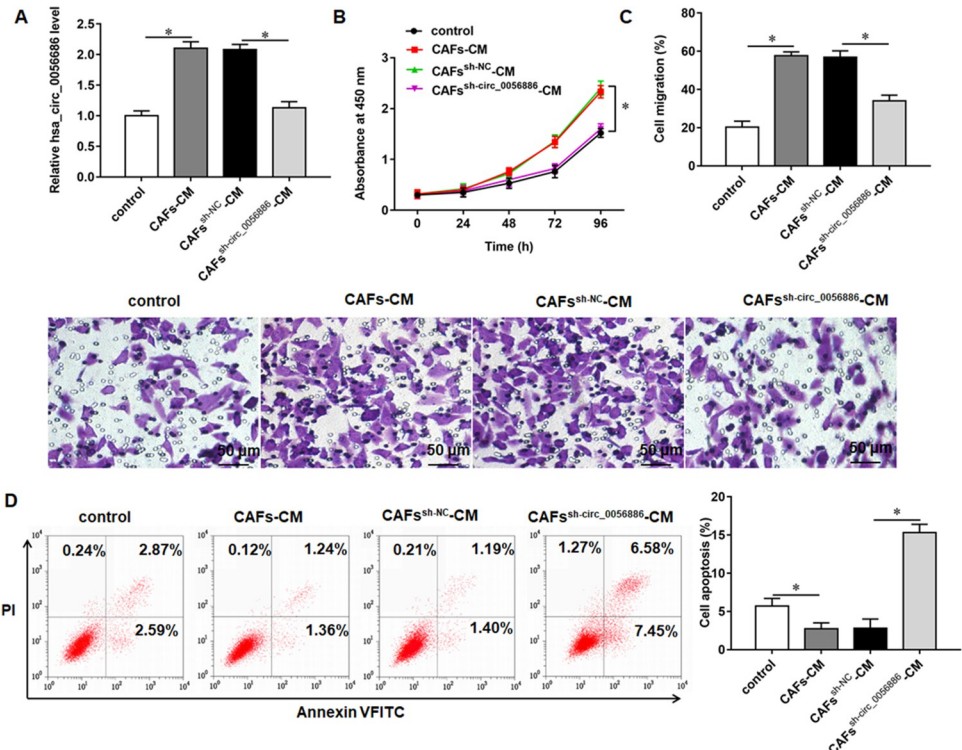

**Fig 3. Effect of knockdown of hsa_circ_0056686 in CAFs on the proliferation, migration and apoptosis of LSMCs.** LSMCs were treated with CAFs-CM or CAFs^sh-circ_0056686-CM. (A) The expression of hsa_circ_0056686 in LSMCs was examined by RT-qPCR assay. (B) The proliferation ability of LSMCs was detected by CCK-8 assay. (C) The migration ability of LSMCs was detected by transwell assay. (D) The apoptosis ratio of LSMCs was detected by flow cytometry assay. $^*P < 0.05$, *vs* control or CAFs^sh-NC-CM.

## Knockdown of hsa_circ_0056686 facilitated ERS-induced cell apoptosis

We next explored the potential molecular pathways responsible for apoptosis. It is well known that ERS activation could induce cell apoptosis. Thus, we further investigated whether hsa_circ_0056686 silencing induce apoptosis by activating ERS. Using Western blot assays, the expression of the ERS markers glucose regulated protein 78 (GRP78), C/EBP homologous protein (CHOP) and activating transcription factor 6 (ATF6) was analyzed in LSMCs treated with CAFs-CM, CAFs^sh-NC-CM and CAFs^sh-circ_0056686-CM. The results showed that the CAFs-CM inhibited the expression of GRP78, CHOP and ATF6 in LSMCs, whereas CM from CAFs^sh-circ_0056686 remarkably enhanced the expression of GRP78, CHOP and ATF6, suggesting an enhanced ERS response in LSMCs (Fig 4A). Meanwhile, the expression of Blc-2 was elevated and the expression of Cleaved-caspase3 was decreased in LSMCs incubated with CAFs-CM, which could be reversed by CAFs^sh-circ_0056686-CM-mediated ERS activation (Fig 4B).

## Knockdown of hsa_circ_0056686 suppressed extracellular matrix (ECM) accumulation

There is growing evidence that accumulation of ECM is one of the important characteristics of ULM. In order to evaluate whether hsa_circ_0056686 can affect ECM, the expression of collagen type I alpha 1 (COL1A1) and alpha 2 (COL1A2) and collagen type III alpha 1 (COL3A1) was assessed. It was found that CAFs-CM could significantly upregulate the expression of

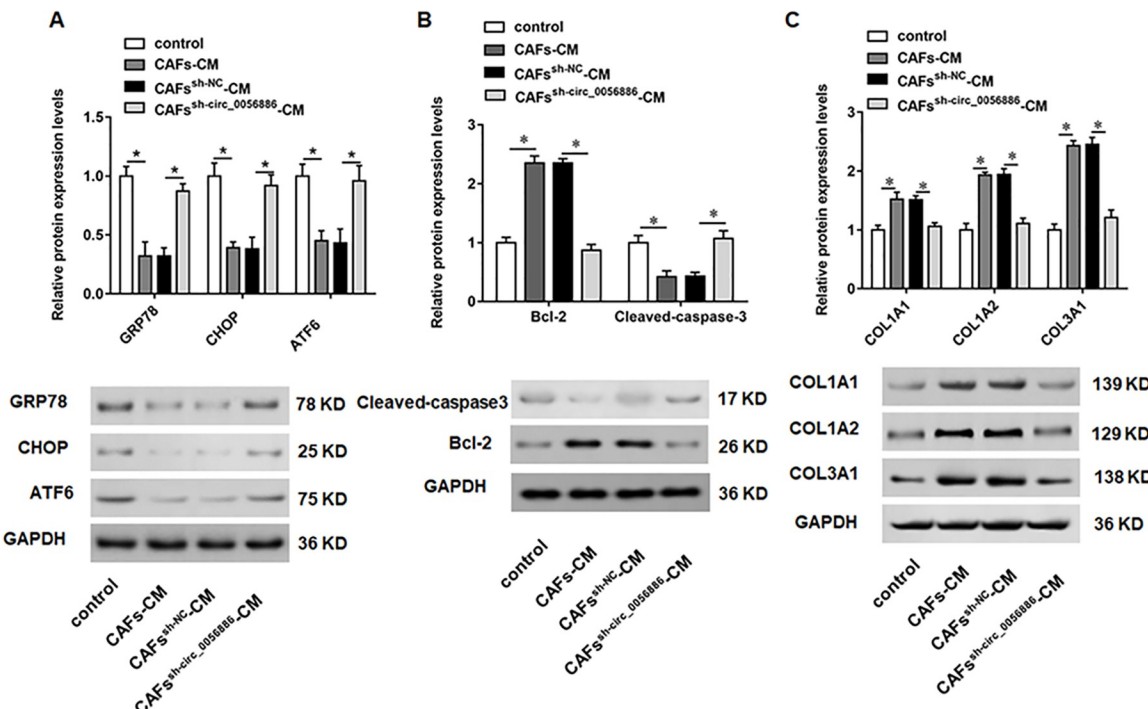

**Fig 4. Effect of knockdown of hsa_circ_0056686 in CAFs on ERS and ECM.** LSMCs were treated with CAFs-CM or CAFs^sh-circ_0056686-CM. (A) The expression of ERS-related proteins GRP78, CHOP and ATF6 was examined. (B) The expression of apoptosis-related proteins Cleaved-caspase3 and Bcl-2 was examined. (C) The expression of ECM-related proteins COL1A1, COL1A2 and COL3A1 was examined. *$P < 0.05$, *vs* control or CAFs^sh-NC-CM.

COL1A1, COL1A2 and COL3A1, while CAFs^sh-circ_0056686-CM led to opposite results (Fig 4C). These results revealed that hsa_circ_0056686 could significantly regulate the accumulation of ECM.

## Knockdown of hsa_circ_0056686 reversed the promoting effect of brefeldin A on ERS

Brefeldin A is a macrolide antibiotic with anti-bacterial and anti-cancer activities, which can promote apoptosis by inducing ERS process. Therefore, brefeldin A, as the ERS inducer, was used to stimulate LSMCs at a final concentration of 10 mg/mL for 24 h. Flow cytometry assay showed that brefeldin A significantly increased apoptosis of LSMCs compared with control, while CAFs^sh-circ_0056686-CM could reverse the inhibitory effect of CAFs-CM on brefeldin A-induced apoptosis (Fig 5A). Meanwhile, the increased levels of GRP78, CHOP and ATF6 caused by brefeldin A were decreased after treatment of LSMCs with CAFs-CM, and CM from CAFs^sh-circ_0056686 restored CAFs-CM-mediated the downregulation of GRP78, CHOP and ATF6 (Fig 5B).

## MiR-515-5p was a direct target of hsa_circ_0056686

To deeply explore the molecular mechanism of hsa_circ_0056686 in ULM, we predicted the downstream target genes of hsa_circ_0056686 through Starbase analysis, and found that hsa_-circ_0056686 contain binding sites for miR-515-5p in 3'UTR (Fig 6A). Moreover, we also found that miR-515-5p was low expressed in ULM tissues (Fig 6B). Subsequently, the results of luciferase reporter gene assay showed that luciferase activity was significantly decreased

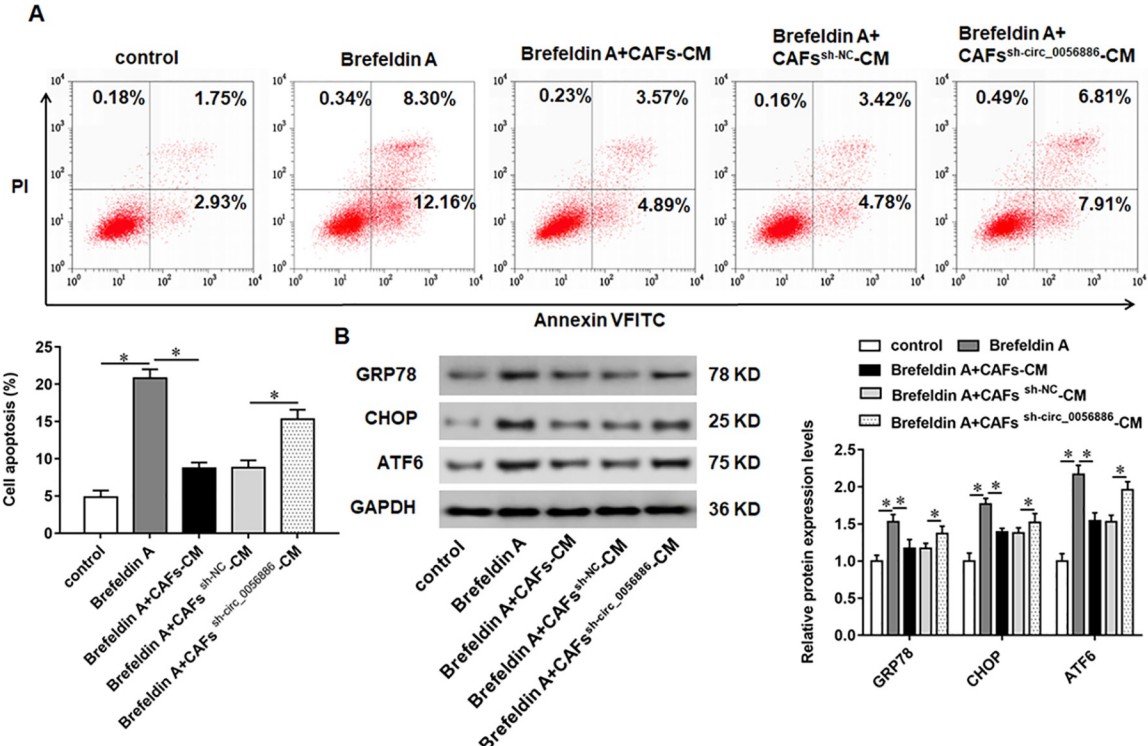

**Fig 5. Knockdown of hsa_circ_0056686 reversed the effect of brefeldin A on ERS.** LSMCs were treated with brefeldin A (10 mg/mL) alone or together with CAFs-CM or CAFs$^{\text{sh-circ\_0056686}}$-CM. (A) The apoptosis ratio of LSMC was detected by flow cytometry assay. (B) The expression of ERS-related proteins GRP78, CHOP and ATF6 was examined. *$P < 0.05$, *vs* control, brefeldin A or brefeldin A + CAFs$^{\text{sh-NC}}$-CM.

upon co-transfection with miR-515-5p mimic and hsa_circ_0056686-wt. No significant difference in luciferase activity was observed after co-transfection with miR-515-5p mimic and hsa_circ_0056686-mut (Fig 6C). In comparison with the bio-probe-NC group, hsa_circ_0056686 was markedly increased in the bio-miR-515-5p-wt group, while no significant difference was observed in the Bio-miR-515-5p-mut group, which was further indicative of a targeting-gene relationship of hsa_circ_0056686 with miR-515-5p (Fig 6D). In addition, it was evident that the expression of miR-515-5p was downregulated or upregulated after overexpressing or silencing hsa_circ_0056686 (Fig 6E). A-Kinase Anchoring Protein 13 (AKAP13) has been reported to be highly expressed in ULM tissues, and we found a conserved binding sequence between miR-515-5p and AKAP13, suggesting that AKAP13 might be a direct regulatory target of miR-515-5p in ULM (Fig 6F).

## Hsa_circ_0056686 exerted its function through miR-515-5p in ULM

To explore the role of miR-515-5p in ULM, CAFs$^{\text{sh-circ\_0056686-CM}}$ treated LSMCs were transfected with miR-515-5p mimic or miR-515-5p inhibitor, respectively. Transwell migration assay and flow cytometry results showed that overexpression of miR-515-5p inhibited cell migration and promoted cell apoptosis compared with CAFs cells transfected with sh-hsa_circ_0056686 (Fig 7A and 7B). In addition, overexpression of miR-515-5p promoted cleaved caspase3, GRP78, CHOP and ATF6 protein expression, inhibited Bcl-2, COL1A1, COL1A2 and COL1A3 protein expression, and transfection of miR-515-5p inhibitor to cells had the opposite effect as transfection of miR-515-5p mimic (Fig 7C and 7D).

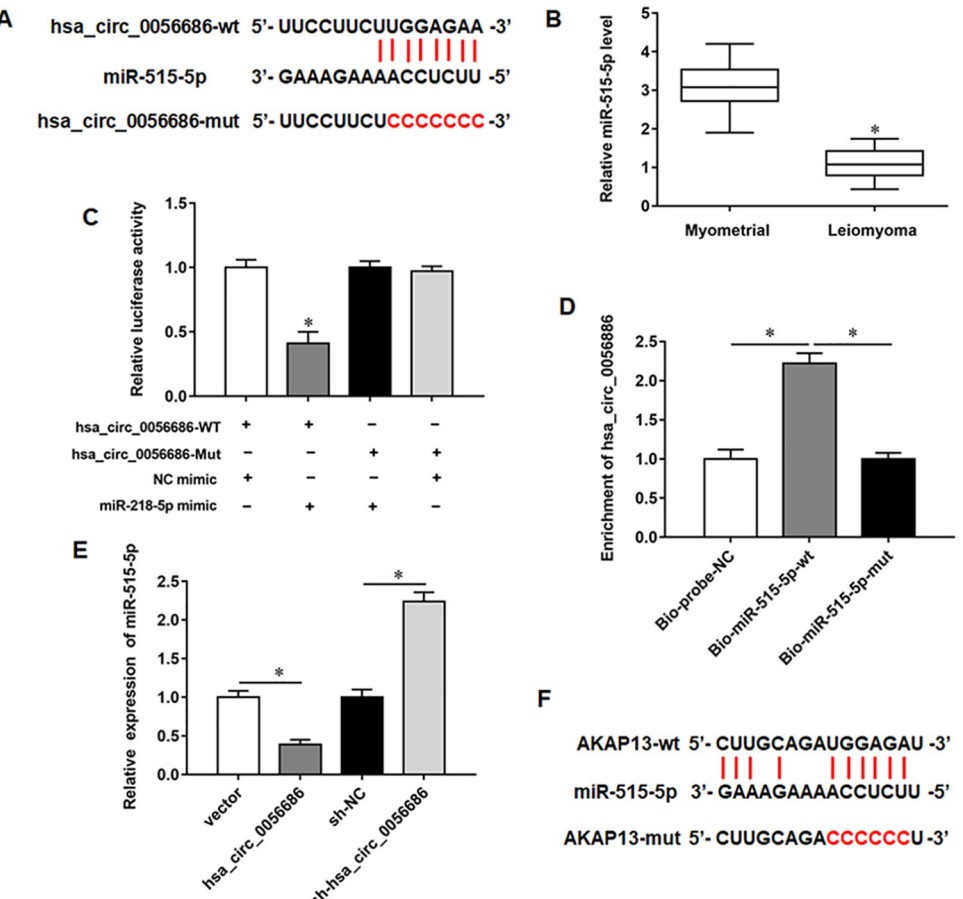

**Fig 6. MiR-515-5p was a direct target of hsa_circ_0056686.** (A) The sequences of hsa_circ_0056686 containing the miR-515-5p binding sites or mutant binding sites were showed. (B) (n = 30) Relative miR-515-5p expression level in ULM tissues and normal myometrial tissues. (C) The luciferase activity was tested after transfecting hsa_circ_0056686-wt and hsa_circ_0056686-mut. (D) RNA pull down assay was performed to detect the enrichment of hsa_circ_0056686 in the Bio-miR-515-5p-wt group and Bio-probe-NC group. (E) The expression of miR-515-5p was detected after overexpressing or silencing hsa_circ_0056686. (F) The sequences of miR-515-5p containing the AKAP13 binding sites or mutant binding sites were showed. *$P < 0.05$, *vs* normal myometrial tissues (B), NC mimic (C), Bio-probe-NC (D), vector or sh-NC (E).

## Discussion

CAFs are initially described as a heterogeneous subgroup of fibroblasts, activated by tumor cells and displayed special markers, which can be considered as prognostic biomarkers in cancers. It has been proofed by extensive researches that CAFs are key players in multiple cancer, including breast cancer and ovarian cancer. The dysregulation of lncRNAs and miRNAs in CAFs has been reported to be involved in tumor growth and progression, nevertheless, the function of circRNAs in CAFs of tumor microenvironment remains ambiguous. Hsa_-circ_0056686 is an exon circRNA formed by splicing of the KIF5C mRNA precursor. KIF5C was reported to be involved in the progression of ULM and closely related to ERS [15,16]. The present study showed that hsa_circ_0056686 was upregulated in ULM tissues, which was consistent with those reported by Wang et al. suggesting that hsa_circ_0056686 was highly expressed in ULM and was positively related with the leiomyoma size [12]. Also, we found that hsa_circ_0056686 in CAFs of ULM was remarkably increased compared with NFs. Also, CAFs

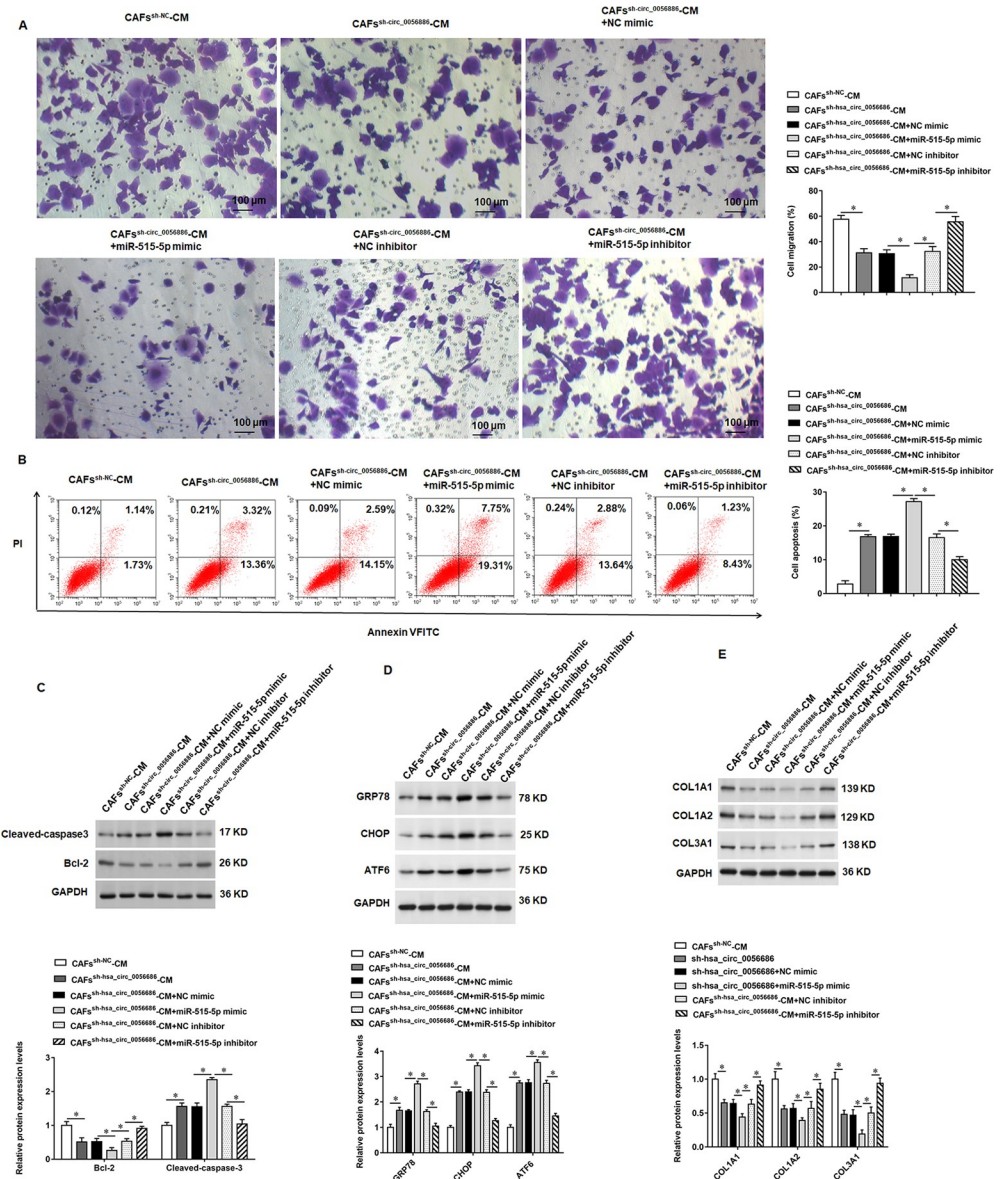

**Fig 7. Hsa_circ_0056686 exerted the tumor-promoting ability of CAFs by directly targeting miR-515-5p.** CAFs<sup>sh-circ_0056686</sup>-CM treated LSMCs were transfected with miR-515-5p mimic or miR-515-5p inhibitor, respectively. (A) The migration ability of LSMCs was detected by Transwell assay. (B) The apoptosis ratio of LSMCs was detected by flow cytometry assay. (C) The expression of apoptosis-related proteins Cleaved-caspase3 and Bcl-2 was examined. (D) The expression of ERS-related proteins GRP78, CHOP and ATF6 was examined. (E) The expression of ECM-related proteins COL1A1, COL1A2 and COL3A1 was examined. *$P < 0.05$, *vs* CAFs<sup>sh-NC</sup>-CM or CAFs<sup>sh-NC</sup>-CM + NC mimic.

could accelerate LMSC proliferation and migration, and inhibit apoptosis. Furthermore, the downregulation of hsa_circ_0056686 in CAFs significantly suppressed proliferation and migration, while induced apoptosis in ULM. ECM mainly contains fibroblasts and COL1A1 and COL3A1 produced by them, affecting the growth of ULM [17]. Myoma cells interact with fibroblasts as well as various growth factors, providing a suitable microenvironment for the formation and growth of leiomyoma [18,19]. Herein, it was detected that CM from CAFs<sup>sh-</sup>

circ_0056686 could significantly decrease the expression of ECM-related proteins, indicating that hsa_circ_0056686 could play an important role in the occurrence of ULM through regulating ECM.

The ERS, mediated by three conservative pathways: IRE1a-XBP-1, PERK-eIF2a, and ATF6, plays a key role in cancer development. When misfolded proteins accumulate due to the damage of proteasome degradation system, the ERS will persist or become severe, and then cell apoptosis is induced [20]. The chop gene transcription can be activated by some major proteins of ERS, including ATF4, ATF6, and XBP-1 [21]. The expression of CHOP and caspases-3 is increased in ERS-induced, meanwhile, Blc-2 signal pathway were activated via CHOP [22]. In the current study, knockdown of hsa_circ_0056686 markedly increased the expression levels of GRP780, CHOP, ATF6 and Cleaved-caspase3, whereas decreased Blc-2 expression. These results suggest that knockdown of hsa_circ_0056686 activated both ERS and downstream apoptosis pathway. In recent years, several studies have found that CAFs are tightly linked to ER stress in tumor progression. Zeng et al. found that curcumin induced apoptosis and cell cycle arrest of CAFs, which were mainly caused by ROS mediated ER stress pathway, and mechanistically, curcumin induced upregulation of ROS via PERK-eIF2α-ATF4 axis triggers ER stress in CAFs [23]. Indeed, in a variety of tumor progressions, CAFs function as tumor promoting cellular activities, and inhibition of the ER stress pathway is an important mechanism of immune evasion in cancer cells. Therefore, we believe that CAFs could inhibit ER stress in tumor cells to some extent, and the two synergistically promote tumor progression. However, the specific regulatory mechanisms between tumor cells, CAFs and ER stress exceed the scope of this study, and still need further in-depth investigation.

At present, circRNAs may function as efficient microRNA (miRNA) sponges to regulate gene expression. Zhang et al. found that miR-515-5p was lower expressed in prostate cancer (PCa), and hsa_circ_0057553 facilitated PCa cell viability, migration, invasion and glycolysis and inhibited apoptosis through miR-515-5p [24]. Besides, Huang et al. revealed that hsa_-circ_0008039 contributed to proliferation, migration and invasion *in vitro* and promoted tumor growth *in vivo* by miR-515-5p/CBX4 axis in breast cancer [25]. Here, we proved that miR-515-5p was a downstream target of hsa_circ_0056686 and downregulated in ULM tissues. In addition, rescue assays verified that CM from CAFs[sh-circ_0056686] could reduce LSMC proliferation, migration and ECM accumulation, and facilitate ERS and apoptosis through upregulating miR-515-5p. Furthermore, we predicted that AKAP13 was a potential target of miR-515-5p through online bioinformatics databases. In a previous study, Ng et al. found that AKAP13 was overexpressed in uterine fibroid tissue [26], which set the stage for our subsequent studies. Another study claimed that AKAP13 expression knockdown suppressed the proliferation and invasion of acute myeloid leukaemia cells [27]. Although it has not been clarified in this study whether miR-515-5p plays a regulatory role on uterine fibroid progression via AKAP13, it will be an important target for our subsequent study.

In summary, this study revealed that hsa_circ_0056686, as a novel oncogene, was upregulated in ULM tissues. And CAFs could secret hsa_circ_0056686 to promote LSMC proliferation, migration, ECM accumulation, and suppress ERS-induced apoptosis by inhibiting miR-515-5p. Consequently, hsa_circ_0056686/miR-515-5p axis might provide a new target in the treatment of ULM.

## Supporting information

**S1 Raw images. The full membrane images of all protein involved in the manuscript.** (PDF)

**S1 Dataset. All data involved in this study.**
(XLSX)

## Author Contributions

**Conceptualization:** Meifang Suo, Airong Zhang.

**Data curation:** Meifang Suo, Zhichen Lin, Dongfang Guo, Airong Zhang.

**Formal analysis:** Meifang Suo, Airong Zhang.

**Funding acquisition:** Meifang Suo, Airong Zhang.

**Investigation:** Zhichen Lin, Dongfang Guo, Airong Zhang.

**Methodology:** Meifang Suo, Airong Zhang.

**Project administration:** Meifang Suo, Airong Zhang.

**Resources:** Meifang Suo, Zhichen Lin, Dongfang Guo, Airong Zhang.

**Supervision:** Airong Zhang.

**Writing – original draft:** Meifang Suo, Zhichen Lin, Dongfang Guo, Airong Zhang.

**Writing – review & editing:** Meifang Suo, Zhichen Lin, Dongfang Guo, Airong Zhang.

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
