## [Decision Letter · Decision Letter 0]

24 Aug 2021

PONE-D-21-23535

Hsa_circ_0056686, derived from cancer-associated fibroblasts, promotes cell proliferation and suppresses apoptosis in uterine leiomyoma through inhibiting endoplasmic reticulum stress

PLOS ONE

Dear Dr. Zhang

Thank you for submitting your manuscript to PLOS ONE. After careful consideration, we feel that it has merit but does not fully meet PLOS ONE’s publication criteria as it currently stands. Therefore, we invite you to submit a revised version of the manuscript that addresses the points raised during the review process.

Although the reviewer feels that the study is potentially interesting, the reviewer requests the authors to address several concerns before publication. In addition, I also have several concerns. Specifically, the authors need to show the percentages of cells in each quadrant of the flow cytometry in Fig. 1d, 3d, 5b, 7a. Ideally, the authors should show the mean +/- SD of triplicate samples and determine the statistical significance of percentages of apoptotic cells in each condition. It is unclear why LSMCs spontaneously develop ER stress and undergo apoptosis. Moreover, it is also unclear why the conditioned medium of CAFs attenuates the expression of GPR78, CHOP, and ATF6 in LSMCs. To investigate the mechanisms underlying these issues may beyond the scope of the present study, however, the authors should discuss these points in more detail.

We look forward to receiving your revised manuscript.

Kind regards,

Hiroyasu Nakano, M.D., Ph.D.

Academic Editor

PLOS ONE

Journal Requirements:

PONE-D-21-23535

"NO

Conceptualization: Meifang, Airong Zhang.

Data curation: Meifang, Zhichen Lin, Dongfang Guo, Airong Zhang.

Formal analysis: Meifang, Airong Zhang.

Funding acquisition: Meifang, Airong Zhang.

Methodology: Meifang, Airong Zhang.

Investigation: Zhichen Lin, Dongfang Guo, Airong Zhang.

Project administration: Meifang, Airong Zhang.

Resources: Meifang, Zhichen Lin, Dongfang Guo, Airong Zhang.

Supervision: Airong Zhang.

Writing - original draft: Meifang, Zhichen Lin, Dongfang Guo, Airong Zhang.

Writing - original draft: Meifang, Zhichen Lin, Dongfang Guo, Airong Zhang." 

Reviewers' comments:

Reviewer's Responses to Questions

**Comments to the Author**

1. Is the manuscript technically sound, and do the data support the conclusions?

Reviewer #1: Yes

2. Has the statistical analysis been performed appropriately and rigorously? 

Reviewer #1: Yes

3. Have the authors made all data underlying the findings in their manuscript fully available?

Reviewer #1: Yes

4. Is the manuscript presented in an intelligible fashion and written in standard English?

Reviewer #1: Yes

5. Review Comments to the Author

Reviewer #1: The authors demonstrated that hsa_circ_0056686 was up-regulated in CAFs around uterine leiomyoma, and influenced the CAFs-mediated proliferation, migration and apoptosis of ULM cells. They also showed that hsa_circ_0056686 directly bound with miR-515-5p and this bounding contributed to the influence on the CAFs-mediated proliferation, migration, and apoptosis.

This study is potentially interesting, however, some issues need to be addressed.

Listed below are my specific concerns and suggestions

1. The effect of hsa_circ_0056686 on the CAFs-mediated proliferation, migration and apoptosis of ULM cells seems to be totally dependent on miR-515-5p. The data examining effects of the miR-515-5p antagomir on the CAFs-mediated proliferation, migration and apoptosis would strengthen the author’s claim.

2. The authors proposed AKAP13 as a possible target of miR-515-5p. Are there any previous reports or data that demonstrated the contribution of AKAP13 to proliferation, migration and apoptosis?

3. In figure 4B, the authors should demonstrate the western blotting data using anti-‘cleaved’ caspase 3 antibody.

4. Add a molecular weight marker to all western blot images in this manuscript.

5. More thorough explanations and important details on the experimental procedures is missing throughout the paper. In particular, effort should be performed for clarifying method to detect the protein amounts of collagens and the expression level of hsa_circ_0056686.

6. PLOS authors have the option to publish the peer review history of their article (what does this mean?). If published, this will include your full peer review and any attached files.

Reviewer #1: No

---

## [Author Response · Author response to Decision Letter 0]

18 Jan 2022

Dear Editor,

Thanks for your letter. We highly appreciate the valuable comments raised by the reviewers on our manuscript entitled " Hsa_circ_0056686, derived from cancer-associated fibroblasts, promotes cell proliferation and suppresses apoptosis in uterine leiomyoma through inhibiting endoplasmic reticulum stress " (ID: PONE-D-21-23535). We have improved our manuscript according to the comments, and the amendments are highlighted in red in the revised manuscript. Point to point responses are listed below. We would like to resubmit our manuscript for your kind consideration.

Thanks again.

Yours sincerely,

Airong Zhang

Department of Medical Laboratory, Zhumadian Central Hospital, Zhumadian, China.

Email: airongz@21cn.com

Although the reviewer feels that the study is potentially interesting, the reviewer requests the authors to address several concerns before publication. In addition, I also have several concerns. Specifically, the authors need to show the percentages of cells in each quadrant of the flow cytometry in Fig. 1d, 3d, 5b, 7a. Ideally, the authors should show the mean +/- SD of triplicate samples and determine the statistical significance of percentages of apoptotic cells in each condition. It is unclear why LSMCs spontaneously develop ER stress and undergo apoptosis. Moreover, it is also unclear why the conditioned medium of CAFs attenuates the expression of GPR78, CHOP, and ATF6 in LSMCs. To investigate the mechanisms underlying these issues may beyond the scope of the present study, however, the authors should discuss these points in more detail. 

Response: Good comments. We added the percentage of cells in each quadrant of flow cytometry in figs 1D, 3D, 5B, 7A.

In this study，the data from at least three replicate experiments were presented as mean ± SD. We modified the poorly described sentences and added more details of statistical analysis in the materials and methods section (Statistical Analysis).

As we know, ER stress signaling in the tumor microenvironment and cancer associated fibroblasts (CAFs) play important roles in tumor progression, and common drivers of ER stress are often low oxygen, nutrients, low pH, and reactive oxygen species accumulation. We fully understand the reviewer′s concerns. In fact, there is some degree of ER stress and apoptosis in cultured cells without any stress (maybe culture in vitro itself is a “stress factor”), and similar results have been reported in previous studies (including but not just Ref.1 & Ref.2).

Ref.1: Tumour suppressor candidate 3 inhibits biological function and increases endoplasmic reticulum stress of melanoma cells WM451 by regulating AKT/GSK3-β/β-catenin pathway. doi: 10.1002/cbf.3515.

Ref.2: MSRB3 promotes the progression of clear cell renal cell carcinoma via regulating endoplasmic reticulum stress. doi: 10.1016/j.prp.2019.152780.

In recent years, several studies have found that CAFs are tightly linked to ER stress in tumor progression. Zeng et al. found that curcumin induced apoptosis and cell cycle arrest of CAFs, which were mainly caused by ROS mediated ER stress pathway, and mechanistically, curcumin induced upregulation of ROS via PERK-eIF2α-ATF4 axis triggers ER stress in CAFs (Ref.3). We fully consider the reviewers' concerns. Indeed, in a variety of tumor progressions, CAFs function as tumor promoting cellular activities, and inhibition of the ER stress pathway is an important mechanism of immune evasion in cancer cells. Therefore, we believe that CAFs could inhibit ER stress in tumor cells to some extent, and the two synergistically promote tumor progression. However, the specific regulatory mechanisms between tumor cells, CAFs and ER stress exceed the scope of this study, and still need further in-depth investigation. We added new literature in the discussion section and discussed the limitations of this study. 

Ref.3: Curcumin promotes cancer-associated fibroblasts apoptosis via ROS-mediated endoplasmic reticulum stress. doi: 10.1016/j.abb.2020.108613.

Response: Thanks. We collated and revised funding information for this study.

Reviewers' comments:

Reviewer's Responses to Questions

Comments to the Author

1. Is the manuscript technically sound, and do the data support the conclusions?

Reviewer #1: Yes

2. Has the statistical analysis been performed appropriately and rigorously?

Reviewer #1: Yes

3. Have the authors made all data underlying the findings in their manuscript fully available?

The PLOS Data policy requires authors to make all data underlying the findings described in their manuscript fully available without restriction, with rare exception (please refer to the Data Availability Statement in the manuscript PDF file). The data should be provided as part of the manuscript or its supporting information, or deposited to a public repository. For example, in addition to summary statistics, the data points behind means, medians and variance measures should be available. If there are restrictions on publicly sharing data-e.g., participant privacy or use of data from a third party—those must be specified.

Reviewer #1: Yes

4. Is the manuscript presented in an intelligible fashion and written in standard English?

Reviewer #1: Yes

5. Review Comments to the Author

Reviewer #1: The authors demonstrated that hsa_circ_0056686 was up-regulated in CAFs around uterine leiomyoma, and influenced the CAFs-mediated proliferation, migration and apoptosis of ULM cells. They also showed that hsa_circ_0056686 directly bound with miR-515-5p and this bounding contributed to the influence on the CAFs-mediated proliferation, migration, and apoptosis.

This study is potentially interesting; however, some issues need to be addressed.

Listed below are my specific concerns and suggestions:

1. The effect of hsa_circ_0056686 on the CAFs-mediated proliferation, migration and apoptosis of ULM cells seems to be totally dependent on miR-515-5p. The data examining effects of the miR-515-5p antagomir on the CAFs-mediated proliferation, migration and apoptosis would strengthen the author’s claim. 

Response: Good comments. The miR-515-5p mimic and miR-515-5p inhibitor were transfected into ULM cells mediated by CAFs, respectively. The results showed that overexpression of miR-515-5p inhibited cell migration and promoted cell apoptosis compared with CAFs cells transfected with sh-hsa_circ_0056686. In addition, overexpression of miR-515-5p promoted cleaved caspase3, GRP78, CHOP and ATF6 protein expression, inhibited Bcl-2, COL1A1, COL1A2 and COL1A3 protein expression, and transfection of miR-515-5p inhibitor to cells had the opposite effect as transfection of miR-515-5p mimic. The specific results were presented in Fig. 7 and results 7.

2. The authors proposed AKAP13 as a possible target of miR-515-5p. Are there any previous reports or data that demonstrated the contribution of AKAP13 to proliferation, migration and apoptosis? 

Response: Good comments. In this study, we predicted that AKAP13 was a potential target of miR-515-5p through online bioinformatics databases, and although it has not been clarified in this study whether miR-515-5p plays a regulatory role on uterine fibroid progression via AKAP13, it will be an important target for our subsequent study. In a previous study, Ng et al. found that AKAP13 was overexpressed in uterine fibroid tissue (Ref 1), which set the stage for our subsequent studies. In addition, another study claimed that AKAP13 expression knockdown suppressed the proliferation and invasion of acute myeloid leukaemia cells (Ref 2). We added the corresponding literature in the discussion section.

Ref 1: A-Kinase Anchoring Protein 13 (AKAP13) Augments Progesterone Signaling in Uterine Fibroid Cells. J Clin Endocrinol Metab. doi: 10.1210/jc.2018-01216.

Ref 2: Long non-coding RNA ZNF667-AS1 knockdown curbs liver metastasis in acute myeloid leukemia by regulating the microRNA-206/AKAP13 Axis. doi: 10.2147/CMAR.S269258.

3. In figure 4B, the authors should demonstrate the western blotting data using anti-‘cleaved’ caspase 3 antibody. 

Response: Thanks. In fact, the antibody used in the experiments was a cleaved caspase 3 antibody (in China, we verbally called it caspase 3 as brief), and we corrected its name in Figure 4B.

4. Add a molecular weight marker to all western blot images in this manuscript. 

Response: Thanks. We labeled the molecular weights of the proteins in all Western blot images.

5. More thorough explanations and important details on the experimental procedures is missing throughout the paper. In particular, effort should be performed for clarifying method to detect the protein amounts of collagens and the expression level of hsa_circ_0056686. 

Response: Thanks. We carefully checked and revised the manuscript and added more details of experimental procedures in the materials and methods section.

6. PLOS authors have the option to publish the peer review history of their article (what does this mean?). If published, this will include your full peer review and any attached files.

Do you want your identity to be public for this peer review? For information about this choice, including consent withdrawal, please see our Privacy Policy.

Reviewer #1: No

We appreciate for Editors/Reviewers’ warm work earnestly, and hope that the correction will meet with approval.

Once again, thank you very much for your comments and suggestions.

---

## [Decision Letter · Decision Letter 1]

2 Mar 2022

PONE-D-21-23535R1Hsa_circ_0056686, derived from cancer-associated fibroblasts, promotes cell proliferation and suppresses apoptosis in uterine leiomyoma through inhibiting endoplasmic reticulum stressPLOS ONE

Dear Dr. Zhang

Thank you for submitting your manuscript to PLOS ONE. After careful consideration, we feel that it has merit but does not fully meet PLOS ONE’s publication criteria as it currently stands. Therefore, we invite you to submit a revised version of the manuscript that addresses the points raised during the review process.

The reviewer has been satisfied with your revision, however, the reviewer still has minor concerns. Specifically, although the authors added the molecular weight markers in Figures, there is no molecular weight markers in Supplementary Figures. Moreover, the order of Figures are not correct; Figure 1 appears to be placed after Figure 7. Thus, I would like to recommend that the authors respond to these points raised by the reviewers. 

We look forward to receiving your revised manuscript.

Kind regards,

Hiroyasu Nakano, M.D., Ph.D.

Academic Editor

PLOS ONE

Journal Requirements:

Reviewers' comments:

Reviewer's Responses to Questions

**Comments to the Author**

1. If the authors have adequately addressed your comments raised in a previous round of review and you feel that this manuscript is now acceptable for publication, you may indicate that here to bypass the “Comments to the Author” section, enter your conflict of interest statement in the “Confidential to Editor” section, and submit your "Accept" recommendation.

Reviewer #1: (No Response)

2. Is the manuscript technically sound, and do the data support the conclusions?

Reviewer #1: Yes

3. Has the statistical analysis been performed appropriately and rigorously? 

Reviewer #1: Yes

4. Have the authors made all data underlying the findings in their manuscript fully available?

Reviewer #1: Yes

5. Is the manuscript presented in an intelligible fashion and written in standard English?

Reviewer #1: Yes

6. Review Comments to the Author

Reviewer #1: The authors did not answer my comment #4 'Add a molecular weight marker to all western blot images.'

7. PLOS authors have the option to publish the peer review history of their article (what does this mean?). If published, this will include your full peer review and any attached files.

Reviewer #1: No

---

## [Author Response · Author response to Decision Letter 1]

17 Mar 2022

Dear Editor,

Thanks for your letter. We highly appreciate the valuable comments raised by the reviewers on our manuscript entitled " Hsa_circ_0056686, derived from cancer-associated fibroblasts, promotes cell proliferation and suppresses apoptosis in uterine leiomyoma through inhibiting endoplasmic reticulum stress " (ID: PONE-D-21-23535R1). We have improved our manuscript according to the comments, and the amendments are highlighted in red in the revised manuscript. Point to point responses are listed below. We would like to resubmit our manuscript for your kind consideration.

Thanks again.

Yours sincerely,

Airong Zhang

Department of Medical Laboratory, Zhumadian Central Hospital, Zhumadian, China.

Email: airongz@21cn.com

The reviewer has been satisfied with your revision, however, the reviewer still has minor concerns. Specifically, although the authors added the molecular weight markers in Figures, there is no molecular weight markers in Supplementary Figures. Moreover, the order of Figures are not correct; Figure 1 appears to be placed after Figure 7. Thus, I would like to recommend that the authors respond to these points raised by the reviewers. 

Response: Sorry for our carelessness. Fist, we added the molecular weight markers in Supplementary Figures. In addition, we have adapted the order of figures 1 and 7 when uploading our latest revised manuscript. 

"No

Conceptualization: Meifang, Airong Zhang.

Data curation: Meifang, Zhichen Lin, Dongfang Guo, Airong Zhang.

Formal analysis: Meifang, Airong Zhang.

Funding acquisition: Meifang, Airong Zhang.

Methodology: Meifang, Airong Zhang.

Investigation: Zhichen Lin, Dongfang Guo, Airong Zhang.

Project administration: Meifang, Airong Zhang.

Resources: Meifang, Zhichen Lin, Dongfang Guo, Airong Zhang.

Supervision: Airong Zhang.

Writing - original draft: Meifang, Zhichen Lin, Dongfang Guo, Airong Zhang.

Writing - original draft: Meifang, Zhichen Lin, Dongfang Guo, Airong Zhang."

Response: Thanks. We collated and revised financial disclosures for this study. Details are as follows, 

“Authors' contributions

Conceptualization: Meifang, Airong Zhang.

Data curation: Meifang, Zhichen Lin, Dongfang Guo, Airong Zhang.

Formal analysis: Meifang, Airong Zhang.

Methodology: Meifang, Airong Zhang.

Investigation: Zhichen Lin, Dongfang Guo, Airong Zhang.

Project administration: Meifang, Airong Zhang.

Resources: Meifang, Zhichen Lin, Dongfang Guo, Airong Zhang.

Supervision: Airong Zhang.

Writing - original draft: Meifang, Zhichen Lin, Dongfang Guo, Airong Zhang.

Writing - original review & editing: Meifang, Zhichen Lin, Dongfang Guo, Airong Zhang.

Funding

The author(s) received no specific funding for this work.” 

Reviewers' comments: 

Reviewer's Responses to Questions

Comments to the Author

1. If the authors have adequately addressed your comments raised in a previous round of review and you feel that this manuscript is now acceptable for publication, you may indicate that here to bypass the “Comments to the Author” section, enter your conflict of interest statement in the “Confidential to Editor” section, and submit your "Accept" recommendation.

Reviewer #1: (No Response)

2. Is the manuscript technically sound, and do the data support the conclusions?

Reviewer #1: Yes

3. Has the statistical analysis been performed appropriately and rigorously?

Reviewer #1: Yes

4. Have the authors made all data underlying the findings in their manuscript fully available?

Reviewer #1: Yes

5. Is the manuscript presented in an intelligible fashion and written in standard English?

Reviewer #1: Yes

6. Review Comments to the Author

Reviewer #1: The authors did not answer my comment #4 'Add a molecular weight marker to all western blot images.'

Response: Sorry for our carelessness. We added the molecular weight markers in Supplementary Figures of latest revised manuscript. 

7. PLOS authors have the option to publish the peer review history of their article (what does this mean?). If published, this will include your full peer review and any attached files.

Do you want your identity to be public for this peer review? For information about this choice, including consent withdrawal, please see our Privacy Policy.

Reviewer #1: No

We appreciate for Editors/Reviewers’ warm work earnestly, and hope that the correction will meet with approval.

Once again, thank you very much for your comments and suggestions.

---

## [Decision Letter · Decision Letter 2]

21 Mar 2022

Hsa_circ_0056686, derived from cancer-associated fibroblasts, promotes cell proliferation and suppresses apoptosis in uterine leiomyoma through inhibiting endoplasmic reticulum stress

PONE-D-21-23535R2

Dear Dr. Zhang

We’re pleased to inform you that your manuscript has been judged scientifically suitable for publication and will be formally accepted for publication once it meets all outstanding technical requirements.

Kind regards,

Hiroyasu Nakano, M.D., Ph.D.

Academic Editor

PLOS ONE

Additional Editor Comments (optional):

Reviewers' comments:

Reviewer's Responses to Questions

**Comments to the Author**

1. If the authors have adequately addressed your comments raised in a previous round of review and you feel that this manuscript is now acceptable for publication, you may indicate that here to bypass the “Comments to the Author” section, enter your conflict of interest statement in the “Confidential to Editor” section, and submit your "Accept" recommendation.

Reviewer #1: (No Response)

2. Is the manuscript technically sound, and do the data support the conclusions?

Reviewer #1: (No Response)

3. Has the statistical analysis been performed appropriately and rigorously? 

Reviewer #1: (No Response)

4. Have the authors made all data underlying the findings in their manuscript fully available?

Reviewer #1: (No Response)

5. Is the manuscript presented in an intelligible fashion and written in standard English?

Reviewer #1: (No Response)

6. Review Comments to the Author

Reviewer #1: (No Response)

7. PLOS authors have the option to publish the peer review history of their article (what does this mean?). If published, this will include your full peer review and any attached files.

Reviewer #1: No

---

## [Editor Report · Acceptance letter]

29 Mar 2022

PONE-D-21-23535R2 

Hsa_circ_0056686, derived from cancer-associated fibroblasts, promotes cell proliferation and suppresses apoptosis in uterine leiomyoma through inhibiting endoplasmic reticulum stress 

Dear Dr. Zhang:

I'm pleased to inform you that your manuscript has been deemed suitable for publication in PLOS ONE. Congratulations! Your manuscript is now with our production department. 

Kind regards, 

on behalf of

Professor Hiroyasu Nakano 

Academic Editor

PLOS ONE